# GNSS Antenna Pattern Prediction and Placement Optimization: A Prototype Method Using Machine Learning to Aid Complex Electromagnetic Simulations Validated on a Vehicle Model

Franciele Cicconet *, Rui Silva and Paulo M. Mendes *

Center for Microelectromechanical System (CMEMS-UMinho), University of Minho, Campus de Azurém, 4800-058 Guimarães, Portugal
* Correspondence: b12408@cmems.uminho.pt (F.C.); paulo.mendes@dei.uminho.pt (P.M.M.)

**Abstract:** An antenna's radiation pattern is dependent on its geometrical characteristics and its antenna's surroundings, materials, and geometries. As such, to predict the antenna's performance in complex environments, such as that of small antennas on large vehicles, it is necessary to obtain a model that represents such a full scenario, so that the simulation may be accomplished in the process of antenna design and placement. Due to the complex and electrically large nature of some electromagnetic problems, the detailed representation, even for a simplified model, may imply a large computational effort, both in terms of time and memory, is needed to perform the simulation. This paper evaluates how machine learning models can be used to mitigate the computational effort required to predict the behavior of antennas requiring complex modeling. It is proposed to start from a more simplified model of the electromagnetic structure to obtain a prediction for the correct solution, without needing to simulate the full structure in every iteration, and to combine this with prediction algorithms to obtain the solution of the full problem. The proposed solution uses convolutional neural networks (U-Net) of a certain accuracy to help with the correct placement of small antennas on autonomous vehicles. The standard approach requires the simulation of a full model at each test position, requiring high computational time and memory. With this new proposal, it is possible to analyze more positions and radiation patterns in a much shorter time, and with less memory, when compared with the solution from the full model. Along with this methodology for each simulation, a Bayesian optimizer is proposed to improve the search process for the best location, leading to a reduction in the required steps. This methodology was applied to support the correct positioning of a GNSS antenna with reference to a set of performance indicators required for autonomous vehicles, but it can be also applied to larger and more complex structures, allowing one to reduce the simulation time of a large electromagnetic structure and the search time for the optimum location.

**Keywords:** antenna placement; radiation pattern; autonomous vehicles; machine learning; Bayesian optimization; U-Net



## 1. Introduction

Antenna placement in large structures is not a novel issue, but is drawing more attention due to the advent of new mobility technologies. In particular, there is a need to place antennas for an affordable price on trucks or cars, which will be used to precisely (sub centimeter precision) compute each vehicle's position with GNSS systems. For solutions using a brute force method, the full simulation model must run at each of the antenna's possible locations, which might lead to very long times (on the week-long scale) taken to find the best position for the antenna. Thus, to avoid brute force localization methodologies, powerful optimization algorithms may be used to find this ideal position and ensure greater accuracy of partial and final results, while obtaining time savings. The need for new solutions arises from the need to frequently redesign vehicles (due to commercial

aesthetic requirements), on which an antenna must be included. That design, which is conditioned by antenna localization, will benefit if it minimizes the need for anechoic-chamber-mediated measurements of full vehicles, a costly and time consuming process. In this way, faster and predictable solutions based on commercially available software must be investigated, which will simplify finding the correct position of a small antenna on a large vehicle. Defining the best position for an antenna on a fixed structure, such as the roof of an autonomous vehicle, requires the implementation, in electromagnetic simulation software, of all the relevant characteristics of the antenna and vehicle, in order to obtain a model as close as possible to the real scenario, and requires the use of a methodology that will determine the correct localization for the antenna while fulfilling the requirements for a given application. In addition to being accurate, such a model should be able to calculate the antenna's radiation characteristics as fast as possible, since the antenna's properties will be recalculated each time the placement methodology needs a new trial position. However, to have optimization algorithms working efficiently, the accuracy of radiation characteristics used for the objective function is mandatory, and as more details of the vehicle and antenna are included in the model, and the more accurate the electromagnetic simulation solvers are, the more time and computational resources will be required for the simulation to converge. Thus, the response that is provided by electromagnetic simulation solvers must reflect, as much as possible, the desired radiation characteristic of the antenna to ensure the effectiveness of the search algorithms, without increasing the computational time significantly. This work proposes a solution that uses a simplified electromagnetic model, implemented using commercial software and machine learning tools, to maintain accuracy in simulations for large electromagnetic bodies and improve computational time, while defining the correct antenna placement.

## 2. Prior Work

Defining the best electromagnetic model and finding the best antenna placement in an autonomous vehicle can be a challenge, due to the variations in antenna behavior induced, for example, by roof materials or the positioning of one or a set of antennas [1,2].

Genetic algorithms were used to optimize the positioning of multiple antennas on a ship's platform in [3]. They reduced the amount of time needed to simulate a position and determined the superiority of one position at the expense of another by comparing the fitness functions. For the case of an antenna, characteristic models were applied to determine the position of a monopole antenna on a metal roof of a vehicle, which achieved the greatest gain in the horizontal plane [4].

Using electromagnetic simulation tools with optimization algorithms has already been investigated. In [5], the position of a simple model of an antenna on a vehicle was examined using a genetic algorithm and with the XFdtd software and a finite difference time domain (FDTD) solver.

Recent progress in development and use of machine learning in electromagnetic problems for antenna design, synthesis and characterization was shown in [6–8]. ML can be applied to reduce computational complexity, allowing faster, real-time operation, and helping to deal with complicated structures problems in cases where the problem requires accurate simulations.

An investigation on the usage of machine learning in antenna design was done in [9]. It was seen that machine learning accelerated the antenna design process while maintaining high accuracy levels in the prediction of antenna behavior.

A neural network method also can help avoid complexity in smart-antenna-array designs, where the behavior is nonlinear, by establishing a relation between the desired radiation patterns and feeding details, such as voltage and spacing in the real antenna array [10]; or it can be used to help with the design of metasurfaces [11].

In [12], a deep neural network was designed to synthesize the radiation pattern of an antenna array. The authors showed that the model exhibits reasonable performance, and the radiation patterns from the outputs were quite similar to the input radiation pattern.

The main contribution of this work is to provide a methodology that allows using an electromagnetic simulator to get radiation pattern as the output of a simplified vehicle model simulation, and then uses a machine learning model to infer the outputs on the remaining antenna placements but with the accuracy of a full model and in a faster way.

The rest of the paper is organized as follows. Section 3 contains the vehicle model preparation. Section 4 contains the machine learning modeling and training, the U-Net configuration, evaluation metrics, and initial tests. Section 5 contains the optimization model. Section 6 contains the previous results, and Section 7 contains the conclusion.

## 3. Vehicle: Model Preparation

To simulate the electromagnetic behavior of an antenna, the biggest challenge is to model the geometry and define the materials of the vehicle and the antenna. The model preparation was performed in CST MWS software and entailed:

- Simplification of the geometry: removing, altering or joining components;
- Geometry healing: an automatic process that removes CAD incongruities;
- Changes in spatial orientation and position: moving or rotating the model according to what is necessary;
- Defining the materials for each element: for each component, defining proper and realistic materials.

Geometry simplification is important, since vehicle models can be extremely complex, especially if they are close replicas of the physical versions. Numerous elements go into making the vehicle model, and a lot of them are too small, too far away from the antenna or composed of materials that do not affect radiation and should not be considered in the simulation. These parts will, however, make the simulation model more complicated if left in. This is a process that requires a thorough analysis of which elements should be removed, and the only way to perform that analysis is through a trial-and-error process, aided by basic radiation knowledge.

Simplifying the vehicle model can also be dependent on the type of simulation and the position of the antenna. For example, in a scenario where the wavelength in question is smaller than the vehicle's dimensions, an antenna placed near the center of the roof will likely not be affected by other geometric factors, so one may not require more than the roof elements in the simulation; on the other hand, an antenna placed near the edges of the roof will require all the closer geometric factors for more realistic results.

Defining the materials of the geometry is also a complex process. Different materials will have different effects on radiation and will add to the complexity of the simulation. For example, any reflective material, such as metal, tends to create a simpler simulation scenario; at the same time, a dielectric material might have almost no impact on the results and still raise the simulation's complexity.

Knowing which materials compose an element is a challenging task. Additionally, even after knowing those materials, it is necessary to know each material's conductivity and tangent of losses for the required frequencies, information which is not always available. Only with this information is it possible to know if the materials have an impact on the antenna's radiation and if it is necessary to include them.

## 4. Convolutional Neural Network (CNN): Modeling and Training

Despite the possible simplifications in the vehicle's geometry and materials, the estimated run-time of a simulation is still high when considering the full model, which is generally more representative of the real scenario. Testing the antenna's behavior on all possible locations of the vehicle's roof would be impractical, if not impossible, so the search for better placement is performed only considering the vehicle roof, thereby losing accuracy but saving computational time.

As a way of not losing that accuracy without spending a lot of time, it is proposed to obtain the antenna's behavior in some specific placements for the complete and simplified models of the vehicle, as shown in Figure 1a, and based on this, use machine learning tools

to predict precise (full model), Figure 1b, radiation patterns given simplified entries (roof model only).

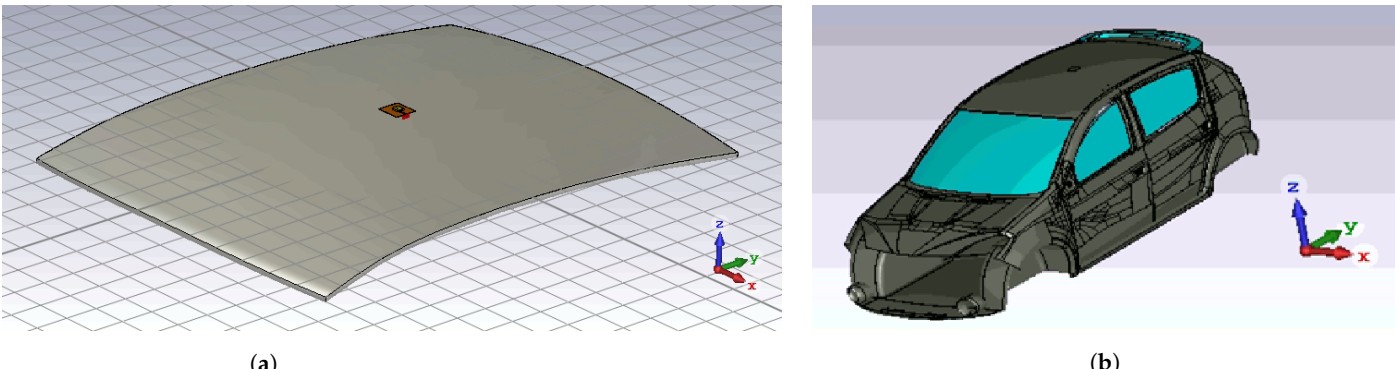

(**a**)                                                                                                  (**b**)

**Figure 1.** Vehicle model preparation in CST MWS software to obtain the data. (**a**) Simplified vehicle model, just the roof. (**b**) Full car model.

The thirteen antenna placements considered to obtain the data for the machine learning model are shown in Figure 2, based on Figure 1a. Thus, using a search optimization algorithm, we can use a simplified model of the vehicle's geometry to obtain faster responses from CST MWS and then use machine learning to approximate the radiation pattern to a more accurate solution, as shown in Figure 3.

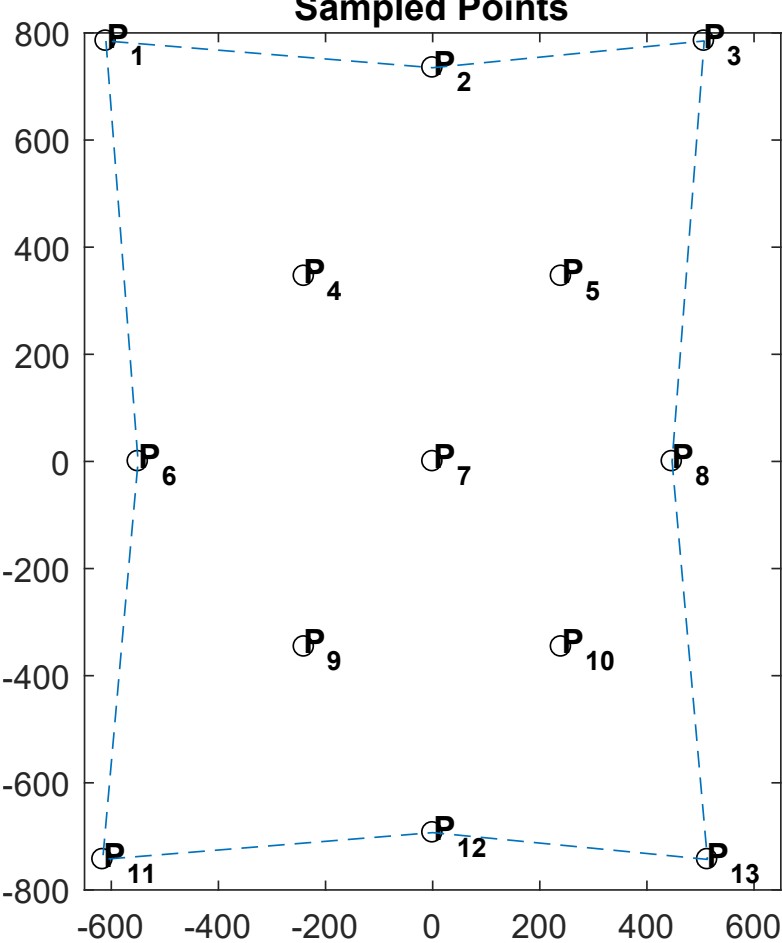

**Figure 2.** Thirteen initial data points (fitting model) of the vehicle roof.

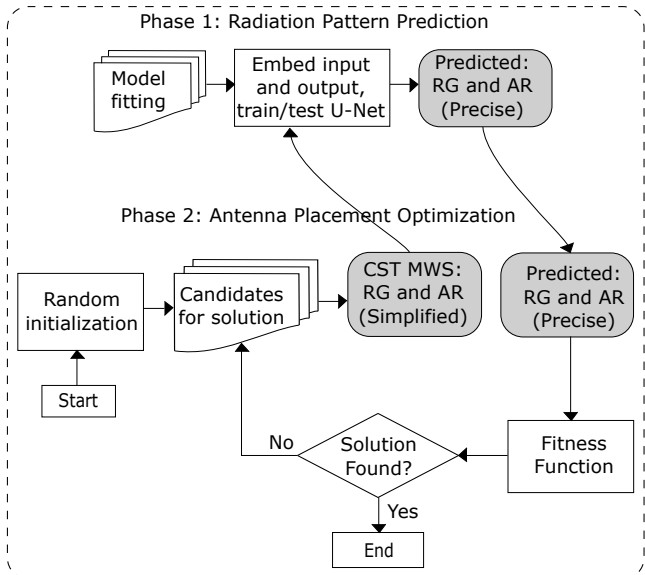

**Figure 3.** Overview of antenna placement optimization with the machine learning model U-Net to predict the radiation pattern.

For the CNN, the antenna placements on the roof of the car (Figure 2) for the full and simplified vehicle models were entered into CST MWS, and the realized gain (RG) and axial ratio (AR) were extracted with MATLAB according to the elevation angle $\theta$, which ranged from $0°$ to $180°$, and azimuth angle $\phi$, which ranged from $0°$ to 360, in units of $1°$. The RG and AR are expressed in *dBi* and *dB*, respectively.

In fitting the model, values obtained from a less accurate electromagnetic model (Figure 1a) were considered as input of the CNN, and a more precise model (Figure 1b) was considered as the output or target of the CNN. It was programmed in Python. The architecture of the model is shown in the next section. As input we have the RG and AR matrices with $\theta \times \phi$ sizes, and as output we have the same size matrices predicted to be used for the search optimization algorithms.

### 4.1. U-Net: Fully Convolutional Neural Network

Convolutional neural networks have existed for a long time and typically are used in developing classification tasks. They have a single class as the output. However, in many visual tasks, such as biomedical ones and those for autonomous vehicles, the output must be an image, where each pixel is classified in a particular class.

Thus, to modify and extend the architecture of the convolutional network to work with a smaller number of images and to perform more accurate image segmentation, U-Net was developed, as shown in Figure 4.

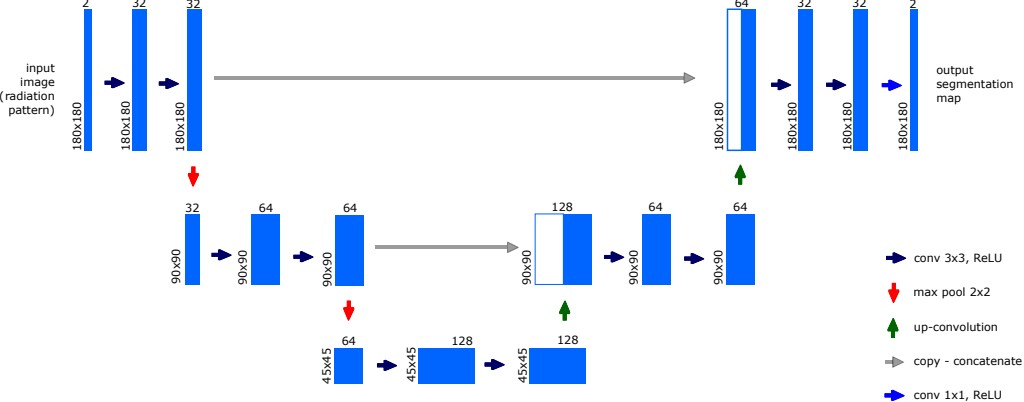

**Figure 4.** U-Net: fully convolutional network.

U-Net has a simple architecture and good performance. It is trained using only convolutions, which can be separated into two paths. The first (left side) is the contraction path (also called encoder) that is used to capture the context of the image. The encoder is made up of convolution layers with a $3 \times 3$ kernel filter, followed by the application of the rectified linear unit (ReLU) activation function, and at the end of each step, a $2 \times 2$ max pooling operation with stride = 2 is performed to reduce by half the size of the image.

Convolution operations may vary depending on the padding and configuration received. The 'stride' is the number of pixels (steps) the kernel advances and is defined as two. As the the resulting image of a convolution can be smaller than the input image size, a padding = 'same' is used to fill the original input before wrapping it so that the output's size is the same as that of the input. Padding can be seen as incrementing the number of pixels added to the edges of the image, as to not lose edge detail.

The second (right side) path is the symmetrical expansion path (also called decoder), which is usually used to find the precise location of an object by using transposed convolutions. It consists of four stages with two types of operations. The first is upsampling, which doubles the size of the image, and the second is upconvolution, which reduces the size of the feature channels by half, the opposite process of convolution.

Due to the loss of information of the border pixels in each layer, a copy of the feature maps is made from left to right. This copy is concatenated with the feature maps resulting from the up-convolution and then undergoes two $3 \times 3$ convolutions, followed by the ReLU activation function and then an up-sampling. Finally, to assign each characteristic vector to the desired number of classes, a $1 \times 1$ convolution is performed on the last layer.

Therefore, U-Net is a fully convolutional network end to end, containing only convolutional layers and no dense layer, which is why it is able to accept images of any size.

### 4.2. Data Augmentation and Evaluation Metrics

When a sufficient amount of data is not available to carry out the training and testing of model, an alternative is to use the data augmentation technique [13]. It consists of generating new data by using simple techniques such as rotation, reflection, displacement and zoom on the original images, thereby increasing the set of initial samples and avoiding over-fitting. This technique contributes to the model being used in cases where it is not possible to obtain a database large enough for training and testing, such as the case here presented: simulation of a full car model in CST MWS, which requires a long time to obtain the radiation pattern data.

All machine learning models minimize or maximize a 'loss function' during the learning, which represents how good a predictive model is at predicting the expected value.

The mean absolute error (MAE), along with MSE, is the most commonly used regression loss function. MAE corresponds to the average of the absolute values of the individual prediction errors in all instances of the test set. The error is defined as the sum of absolute differences between the target values and the predicted values for the instance:

$$\text{MAE} = \frac{\sum_{i=1}^{n} |y_i - y_i^p|}{n} \tag{1}$$

### 4.3. Initial Test

After obtaining the realized gain and axial ratio data of the simplified and precise vehicle models at the points described in Figure 2, the data augmentation techniques were applied.

Then, for each point, the $13 \times 180 \times 360$ realized gain matrices with dimensions $180 \times 360$ were separated into $26 \times 180 \times 180$. As a strategy to increase the data set, 3 rotations of 90 degrees were defined in these 26 matrices, and the number rose to 104. Finally, the order of the elements of all matrices was inverted, giving a set of $208 \times 180 \times 360$. The same was done with the axial ratio matrices.

Figure 5a shows the polar representation of realized gain simulated with CST MWS in the simplified (roof) model and in the precise (full) model; and in Figure 5b, the same precise simulated model (blue line) and that predicted by U-Net (black line).

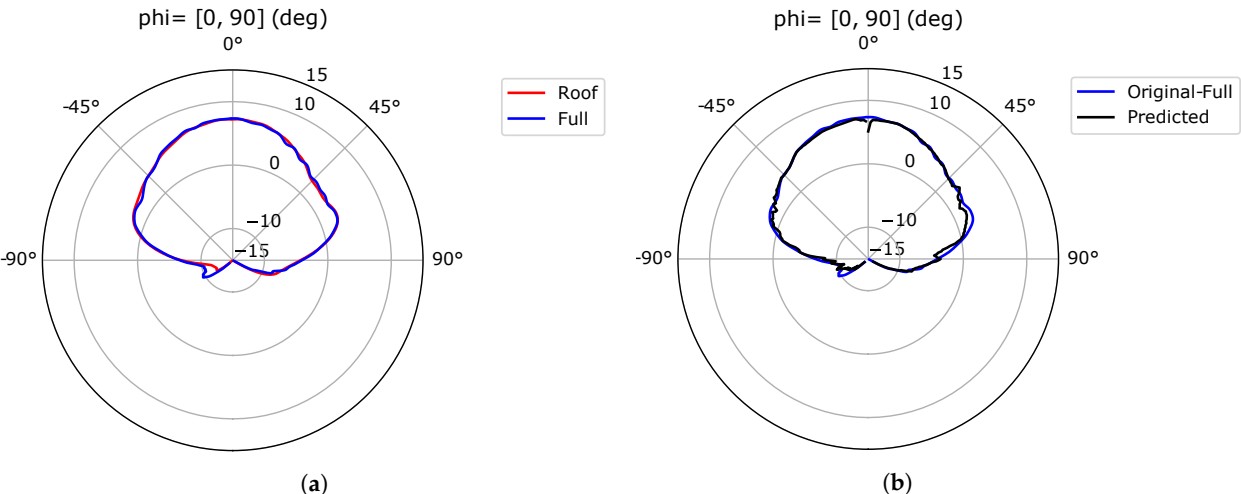

**Figure 5.** Realized gain. (**a**) Roof × full vehicle model simulated by CST. (**b**) Comparison of full car, simulated (CST) and predicted (U-Net).

Figure 6a shows the polar representation of the axial ratio simulated with CST MWS in the simplified and precise vehicle models; and in Figure 6b, the same precise simulated model (blue line) and that predicted by U-Net (black line).

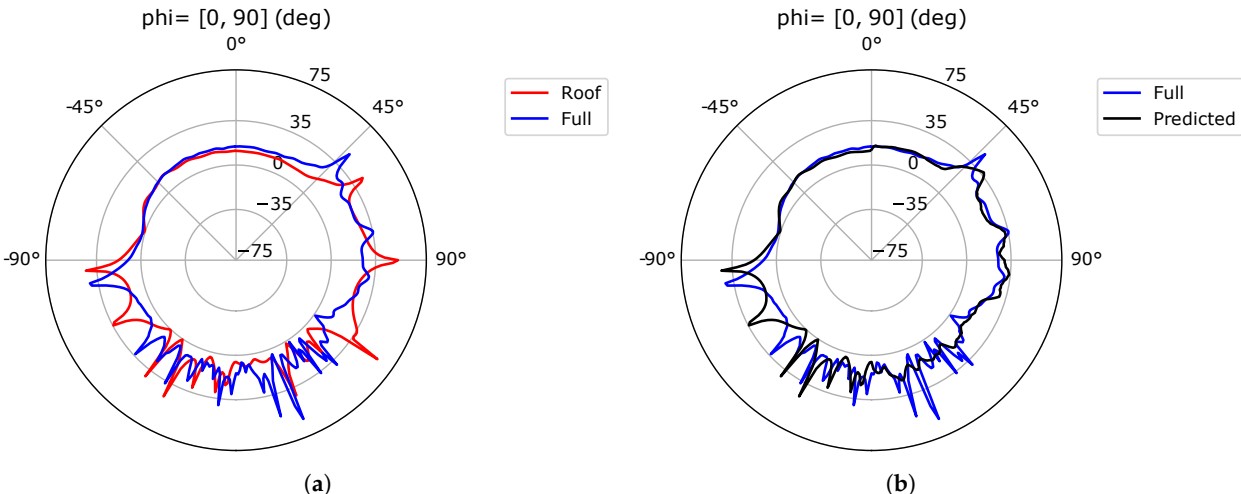

**Figure 6.** Axial ratio. (**a**) Roof × full vehicle model simulated by CST. (**b**) Comparison of full car, simulated (CST) and predicted (U-Net).

In Figure 7 is shown the three dimensional representation of the realized gain of Figure 5 but here for all azimuth angles ($\phi$).

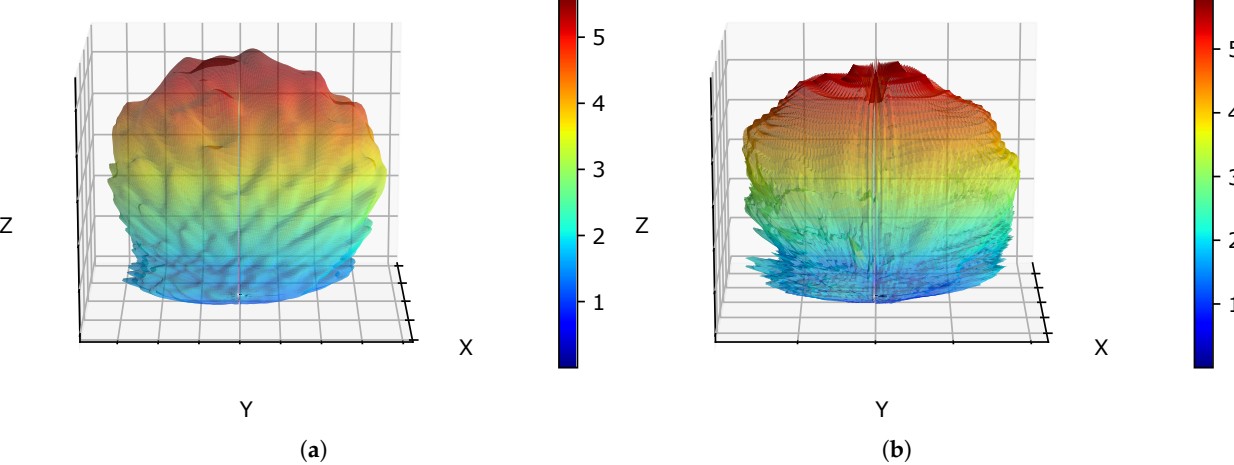

**Figure 7.** Case study. (**a**) Original RG of *P*6 point. (**b**) Predicted RG of *P*6 point.

U-Net's learning process metrics are shown in Figure 8. It was observed that, unlike realized gain, the axial ratio has little correlation between the simplified and the precise models, which affected the prediction and accuracy of the model a little, although very little, as can be seen in Figures 5 and 6.

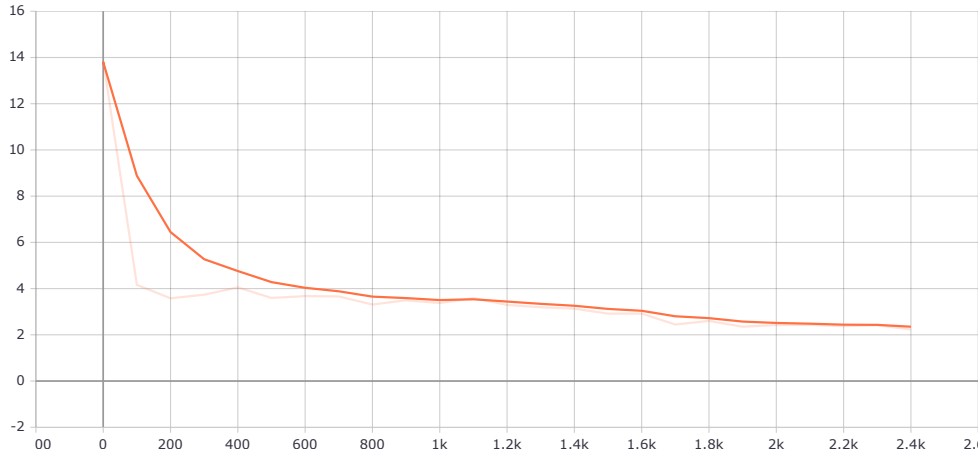

**Figure 8.** Metrics: mean absolute error of the U-Net model vs. steps

In the next section, we report on a Bayesian optimization model being added to the U-Net as a way of finding the best position for an antenna. Thus, a U-Net is intended to replace the electromagnetic simulation of antennas in complete models that require a lot of time, and instead make the prediction from the simplified models. The simplified simulation results are sent to the U-Net, and the prediction results are sent to the Bayesian optimization model for calculation of the objective function, as shown in Figure 3.

## 5. Optimization Model and U-Net

In the second phase, the prediction model of the precise radiation patterns (full vehicle model) is integrated in a Bayesian optimization (BO) model [14]. With it, we can use a less precise antenna model in CST (faster simulation), and with a convolutional neural network, the radiation patterns can be predicted, bringing them closer to more accurate models' results (longer simulation).

The criterion chosen to define the best antenna placement is given by the maximization of the difference between the average realized gain, $\bar{RG}_i$, and the average axial ratio, $\bar{AR}_i$,

below the elevation angle $\theta \leq \theta^{max}$ in the position $i$, for all azimuth angles predicted by neural network and calculated by Equation (2):

$$\min -F(\bar{RG}, \bar{AR}) = \bar{RG}_{\rho_n} - k_1 \bar{AR}_{\rho_n}$$

$$\text{s.t:}$$

$$\begin{cases} BW^{min} \leq BW_{S11_{\rho_n}} \leq BW^{max}, \\ S11_{\rho_n} \leq S11_{max}, \\ n = 1, ..., N; \ \theta = 1^o, ..., \theta^{max}; \ \phi = 1^o, ..., 360^o. \end{cases}$$

(2)

where:

$\bar{RG}_{\rho_n}$ is the realized average gain in position $\rho_n$;
$\bar{AR}_{\rho_n}$ is the average axial ratio in position $\rho_n$;
$BW_{S11_{\rho_n}}$ is the bandwidth in position $\rho_n$;
$S11_{\rho_n}$ is the return loss in position $\rho_n$;
$k_1$ is a constant to normalize the objective function.

The antenna placement problem was solved using BO packages [15,16]. BO builds a probability model of an unknown objective function that would take a long time to evaluate at each point. Thus, through a Gaussian process (GP), a surrogate model with a normal distribution of $i$ samples (with mean and variance defined) is constructed as a probability representation of the objective function [17].

PG is a way to infer a function from samples of its inputs and outputs, also providing a distribution over the outputs. Therefore, when you guess the output of the function at some given point, the GP also tells us the probability of finding it within given ranges [18].

The search for new points of evaluation in the real objective function will be given by an acquisition (or selection) function. This function represents the trade-off between exploring new areas, through the variance of the surrogate function, and taking advantage of known regions, through the mean. Maximizing this function is cheaper from a computational point of view and its maximum point determines the next point $(n + 1)$ to be sampled.

Algorithm 1 shows how Bayesian optimization evaluates the objective function in an initial space and allocates the remainder of a budget of $N$ function evaluations [17].

---

**Algorithm 1** Basic Bayesian optimization pseudo-code.

---

-Place a GP prior function on $F'(\rho)$, by default $\mu(\rho) = 0$ (deterministic)
-Observe $F$ at $n$ position operating on the current prior distribution.
**while** $n \leq N$ **do**
  -Update the posterior probability distribution on $F$ using all available data.
  -Let $\rho_n$ be a maximizer of the acquisition function over $\rho$, where the acquisition function is computed using the current posterior distribution.
  >Get the radiation pattern (realized gain and axial ratio) from the simplified model and send it to U-Net.
  >U-Net: Predict the precises values of radiation pattern (realized gain and axial ratio) and send it back to BO.
  -Observe $F(\rho_n)$.
  -Update the Gaussian Process prior distribution with the new data to produce a posterior (which will become the prior in the next step).
  -Increment $n$.
**end while**
**return** Solution: Return either the point evaluated with the best $F(\rho_n)$, or the point with the largest posterior mean.

---

## 6. Previous Results

To test the optimization algorithm joined with the U-Net, a Delaunay triangulation function [19] was used to provide the realized gain and axial ratio values in the positions $(x, y)$ requested by the Bayesian optimizer for the simplified vehicle model.

Thus, at each search position on the vehicle roof, the Delaunay function triangulates the simplified values simulated in the pre-selected points of Figure 2 and forwards these values to U-Net to predict the precise values (full car). This was the way found to test the joint model (Opt+UNet), since it will still be necessary to have the CST solver communicate with Python to obtain these simulated values in each position.

A solution was found in 2 min 2 s and with 80 iterations. Figure 9 shows the best antenna position with the following information:

$$x, y: -534, -16 \text{ mm}$$
$$F = \bar{rg} - 0.1\bar{ar} = 1.938 \ \bar{rg}: 3.5444 \text{ dBi}$$
$$\bar{ar}: 16.062048 \text{ dB}$$

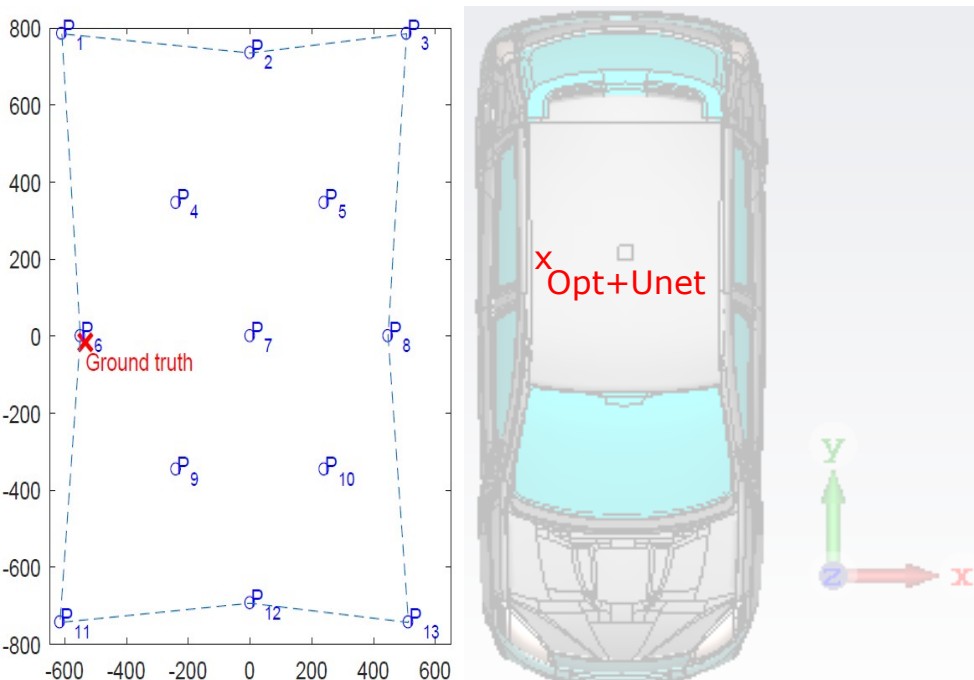

**Figure 9.** Ground truth is in position 6. *F* has the maximum value for all 13 training points, and the Bayesian optimization + U-Net solution found is near this point.

The time spent on the U-Net model training was 1 h 15 min, and each prediction took an average of 0.30 s. The time used to simulate the precise model in CST was at least 24 h at each position.

Thus, clearly the U-Net model is considerably faster on the whole, even considering the simulation time of the simplified model, which was around 1 h and was replaced by triangulation for this initial test.

The Opt+Unet model can also be used to predict return loss and bandwidth, which is left for future work.

## 7. Conclusions

The Opt+Unet model proposed in this paper is a powerful tool to solve magnetic simulation problems, both on small areas such as vehicle roofs and larger areas, which demand a lot of simulation time and resources, especially when searching for the best antenna position.

While the Bayesian optimization model may find the optimal point in the smallest number of objective function calls, the U-Net further helps reduce this time by predicting a precise model from a simplified model faster than CST simulates a precise model directly.

Despite the very promising results, it is suggested that, before applying this methodology to other scenarios, tests be performed to assess the accuracy of the prediction model, to investigate the minimum number of training points and to modify the various hyperparameters of the model. Regarding the Bayesian optimization model, the improvement of a library that handles more complex boundaries could help with improving the methodology's performance. Depending on the required optimization goals, it may be necessary to formulate the problem in terms of axial ratio, return loss and bandwidth to better characterize the desired behavior of an antenna.

## 8. Patents

Patent PCT/IB2022/053002—'Method for Determining the Correct Placement of an Antenna with Radiation Pattern Prediction' was originated from the work reported in this manuscript.

**Author Contributions:** Conceptualization, F.C. and R.S.; Investigation, F.C. and R.S.; Visualization, F.C.; Validation, F.C.; Software, F.C and R.S.; Formal Analysis, F.C. and R.S.; Writing—Original Draft, F.C. and R.S.; Supervision and Coordination, P.M.M.; Antenna Project, P.M.M.; Project Administration, P.M.M. All authors have read and agreed to the published version of the manuscript.

**Funding:** This work is supported by: European Structural and Investment Funds in the FEDER component, through the Operational Competitiveness and Internationalization Programme (COMPETE 2020) (Project number 037902; Funding Reference: POCI-01-0247-FEDER-037902).

**Institutional Review Board Statement:** Not applicable.

**Informed Consent Statement:** Not applicable.

**Data Availability Statement:** Not applicable.

**Conflicts of Interest:** The authors declare no conflict of interest.

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
