# Peer review of "GNSS Antenna Pattern Prediction and Placement Optimization: A Prototype Method Using Machine Learning to Aid Complex Electromagnetic Simulations Validated on a Vehicle Model"

_applsci, doi:10.3390/app13042197_

Round 1

Reviewer 1 Report

The organization of the paper was good but the authors need to enrich some of the revision in the manuscript.

  1. Separate the introduction and related works sections. Provide the major contribution of this research in Introduction section. Also, include the advantages and drawbacks of the existing models and justify, how the proposed model will eliminate the existing issues?

  2. Provide some of the valid points about the antenna radiation pattern and how it was predicted using machine learning model?

  3. Too low references and there is no recent references are cited. Try to include some of the recent references related to the research and cite it in the related work section.

  4. There is no training dataset was provided. Try to include some process working of training and testing values related to the research

  5. Illustrate the results values in numerical form. There are no numerical calculations are available in this research paper. 

Author Response

The organization of the paper was good but the authors need to enrich some of the revision in the manuscript.

  1. Separate the introduction and related works sections. Provide the major contribution of this research in Introduction section. Also, include the advantages and drawbacks of the existing models and justify, how the proposed model will eliminate the existing issues?

Sections are now separated, and content slightly extended.

  1. Provide some of the valid points about the antenna radiation pattern and how it was predicted using machine learning model?

I’m sorry, but I don’t understand what is the meaning by “valid points”. Regarding the prediction, it follows figure 3. First, a set of full model simulations are run, at some selected points with different characteristics (flat surface, corner, or bending point). After that, the process starts with the use of the simplified model. That simplified model is used for iteration at each test position (suggested by the placement algorithm). After having the results from the simplified model, it is used tighter with the U-Net to obtain the accurate results. Those results are than used to verify if the position matches the specified parameters for the objective function.

  1. Too low references and there is no recent references are cited. Try to include some of the recent references related to the research and cite it in the related work section.

We included the references we used to allow the implementation described in the paper. As well some papers that supports this development. Not being a review paper, we don’t think a large number of references make the subject more relevant. This was a need from a company that is working in this subject. This is way a IP was filled along the process.

We agree that are not many recent references on the but that is due to: not many papers are available on the prediction of antenna patterns based on AI methods, and we have reference to papers, which give an overview of the area:

Machine learning in electromagnetics: A review and some perspectives for future research. In Proceedings of the 2019 International Conference on Electromagnetics in Advanced Applications (ICEAA). IEEE, 2019, pp. 1377–1380;

Evaluating Machine Learning &Antenna Placement for Enhanced GNSS Accuracy for CAVs. In Proceedings of the 2019 IEEE Intelligent Vehicles Symposium 315 (IV). IEEE, 2019, pp. 1007–1012.;

A Deep Learning-Based Approach for Radiation Pattern Synthesis of an Array Antenna. IEEE Access 2020

 Machine learning in antenna design: An overview on machine learning concept and algorithms. 330 In Proceedings of the 2019

Anyway, we added extra, more recent, related references:

https://www.mdpi.com/1424-8220/21/8/2765

https://ieeexplore.ieee.org/document/9703477

https://www.intechopen.com/chapters/83306

https://ieeexplore.ieee.org/document/9296770

  1. There is no training dataset was provided. Try to include some process working of training and testing values related to the research

The training dataset publication is not a standard practice for this kind of work. Mostly because it tends to not be useful to share data in some file, in a in-house format made by one student in a company. At the moment we are not able to provide it in a useful way, but we plan to make it available, when data can properly documented and meaningful for others.

  1. Illustrate the results values in numerical form. There are no numerical calculations are available in this research paper.

We consider that the results are presented in the convenient and standard way used by antenna community. That comprises the presentation of radiation patterns, plots showing the curves behavior, and figures showing the results.

Reviewer 2 Report

The paper is well written, and the presentation is nice, but the concept is not particularly novel, as much work has already been done in this area.

Author Response

The paper is well written, and the presentation is nice, but the concept is not particularly novel, as much work has already been done in this area.

Dear reviewer, we appreciate your time taken to review our manuscript, but allow us to, respectfully, partially disagree with your comment.

1 - In fact, placement algorithms have been applied in many applications, and can be reused for other applications, as is this case. However, and maybe we miss the relevant publications, we did not find many demonstrations of using such algorithms for antenna placement in large bodies, or, in particular, GNSS antennas in large vehicles. As we know, the performance of some algorithms will depend on the algorithm itself, and how it’s used. To be aware of an algorithm performance, we have to implement and test it. This is what we did, regarding the placement aspect.

2 – We are aware of works like https://onlinelibrary.wiley.com/doi/pdf/10.1002/mop.10880 or https://www.mdpi.com/1424-8220/21/8/2765 or http://users.auth.gr/antodimi/Journal%20Papers/A%20Novel%20Technique%20for%20the%20Approximation%20of%203-D%20Antenna%20Radiation%20Patterns.pdf, or others, but those methods are looking for different solutions for different approaches to get a faster solution to compute antenna characteristics. In one case, generating FF patterns from near field, or even current distributions, usually arriving at the need to implement more robust boundary conditions. On other cases, deriving an approximation for 3D patterns from a set of measured, or simulated field cuts. Other examples are also available for antenna synthesis, or for antenna input characteristics prediction. We look to obtain results on electrically large models, using reduced models. We made this clearer now in the manuscript.

In our case, we want to predict full-model antenna radiation patterns from a partial electromagnetic model, using commercially available tools, like HFSS, CST, COMSOL, or many others. Of course, if we design our own software, maybe we can do that from near fields, and make the problem more lightweight. But companies usually use commercial software, do not design their own software. And, we may say that the algorithms are available, and that we are “only” using them. But “only” using some idea to solve a technical problem that others did not yet fully, or at all, addressed is usually worthwhile of reporting to the community.

In fact, we took many research hours to be able to reduce the computation time of a full vehicle, from more than a week (each run) to less than a day, or a couple of hours, using a HPC, with graphical boards computing, solid state drives and about 500 GB of memory. We had to come out with a methodology.

We are not aware of solutions that have been particularly used for this case, were we run a small portion of the EM model from an electromagnetically large model, feed it to some AI methodology, and get the FF pattern.

Thank you very much for your time.

Reviewer 3 Report

Review for applsci-2135303

GNSS Antenna Pattern Prediction and Placement Optimization: Using Machine Learning to Aid Complex Electromagnetic Simulations

Franciele Cicconet, Rui Silva, Paulo Mendes

Review:

The authors of this manuscript describe a method that provides a user with the ability to find the best location for antenna placement, where the criterion for best antenna placement is the location where the difference between the average realized gain, RGi, and the average axial ratio, ARi is maximum. For this manuscript, the authors use an example of placing the antenna on the roof of a car.

RG and AR are parameters that define the performance for an antenna at a given location or setting. Antenna designers often chose which parameters they would like to prioritize or maximize. For example, if the desired polarization of an antenna is circular, the ideal value for AR is 0db. These parameters are often easier to analyze when visualized, as done by the authors in Figure 5 and Figure 6 of their manuscript.

The authors describe how these parameters can be calculated using simulations and software such as CST Microwave Studio (CST MWS), and that using simplified models (such as only the roof or with lesser components or combined components) can provide one set of parameters while using less computation time, while using complex models (such as including the full vehicle, or component material information or geometries) can provide another set of parameters that may be “more accurate”, but take a much longer time to compute.

Since the antenna placement approach used by the authors relies on finding the location with the maximum difference between RG and AR, the approach requires the calculation of both these parameters across multiple locations which can require significant computational resources. The authors propose to reduce this requirement by training a deep learning model to “learn” the parameters generated by a full simulation, and use this deep learning model instead of running simulations in CST MWS repeatedly for each location.

I am not satisfied with the manuscript in its current form due to many reasons, with the primary factor being that I am not sure if I could replicate the authors proposed method after reading this manuscript. The secondary factor is that while the authors describe a method, they do not quantify the performance of this method, or compare it to other methods in the literature.

Comments:

(1)   I am not convinced the deep learning model is significantly better than the simplified model. I overlayed Figure 6b and a together, so that all three [“Roof (red)”, “Full (blue)”, and “Predicted (black)”] plots can be seen in the same figure below:

As seen above, the red and the black lines are completely overlapping in the left half of the plot, and only deviate in the right half. The “black” or predicted line seems to fall midway between the blue (“full) and red (“roof”) lines. From the perspective of deep (or machine) learning, we must quantify why any one of these three lines is better than the others, or what the error is.

In the next revision, I suggest the authors plot all three lines in the same figure for a truthful analysis on the value of the proposed method.

(2)   Since the goal of the manuscript is antenna placement, please show what benefits are obtained in the antenna placement process by using a deep learning model, and what are the drawbacks. For example, how different are the locations selected if a simplified model is used, or if the full model is used compared to the proposed UNET parameters. What is the RG and AR at locations selected in these three scenarios? Is the benefit of using this proposed method (saved computation time) worth the reduced performance from not using the full model? How much is the performance different when compared to the simplified model? Showing multiple examples would be helpful.

Author Response

Comments:

(1)   I am not convinced the deep learning model is significantly better than the simplified model. I overlayed Figure 6b and a together, so that all three [“Roof (red)”, “Full (blue)”, and “Predicted (black)”] plots can be seen in the same figure below:

As seen above, the red and the black lines are completely overlapping in the left half of the plot, and only deviate in the right half. The “black” or predicted line seems to fall midway between the blue (“full) and red (“roof”) lines. From the perspective of deep (or machine) learning, we must quantify why any one of these three lines is better than the others, or what the error is.

In the next revision, I suggest the authors plot all three lines in the same figure for a truthful analysis on the value of the proposed method.

Thank you for your analysis. In fact, we decided to use two plots, not for hiding anything, but to not mix to many lines. We would like to keep it like that, which makes it easier to use it when printed. We agree that in this case, the deep learning is not better in all quadrants, but also not worst. And is better in the range [0-90º] and [90-135º]. This means that almost half the radiation diagram is improved.

(2)   Since the goal of the manuscript is antenna placement, please show what benefits are obtained in the antenna placement process by using a deep learning model, and what are the drawbacks. For example, how different are the locations selected if a simplified model is used, or if the full model is used compared to the proposed UNET parameters. What is the RG and AR at locations selected in these three scenarios? Is the benefit of using this proposed method (saved computation time) worth the reduced performance from not using the full model? How much is the performance different when compared to the simplified model? Showing multiple examples would be helpful.

The main benefit is time. Each simulation with the full model (for each candidate position) takes between one and two weeks (depends on how much we strip the vehicle elements). Depending on the simplification used, it was possible to make all simulations to get a position in less than one day. In some cases, in half day. The full model would take one or two months. This worth some penalty in accuracy. Actually, with this approach, we may select two or three candidate positions, and perform the full model simulation, to make the final decision, or perform anechoic chamber measurements.

The performance when using the simplified, sometimes, is catastrophic. Because it depends the path the localization algorithm uses. If it selects a path more on the middle of the roof, the reduced model follows it very well. But if it decides to go (or if it is required by design) around the roof, near the edges or corners, the start to become more different. And sometimes, since antenna characteristics from simplified model are largely different from reality (I would say wrong), we may stop at a completely wrong position. It will not be a question on how far it is from the correct position. It will stop at a completely wrong place.

Hope it is clearer now, and why such plot are not present in the paper. They will simply would look random positions.

Round 2

Reviewer 3 Report

Authors Response #1

Thank you for your analysis. In fact, we decided to use two plots, not for hiding anything, but to not mix to many lines. We would like to keep it like that, which makes it easier to use it when printed. We agree that in this case, the deep learning is not better in all quadrants, but also not worst. And is better in the range [0-90º] and [90-135º]. This means that almost half the radiation diagram is improved

Reviewers Response:

In my opinion, half the radiation diagram improving for one sample is not indicative of the performance of the method as a whole. The choice of presenting the method or the results is inconsistent with contemporary Machine learning literature.

Authors Response #2:

The main benefit is time. Each simulation with the full model (for each candidate position) takes between one and two weeks (depends on how much we strip the vehicle elements). Depending on the simplification used, it was possible to make all simulations to get a position in less than one day. In some cases, in half day. The full model would take one or two months. This worth some penalty in accuracy. Actually, with this approach, we may select two or three candidate positions, and perform the full model simulation, to make the final decision, or perform anechoic chamber measurements.

The performance when using the simplified, sometimes, is catastrophic. Because it depends the path the localization algorithm uses. If it selects a path more on the middle of the roof, the reduced model follows it very well. But if it decides to go (or if it is required by design) around the roof, near the edges or corners, the start to become more different. And sometimes, since antenna characteristics from simplified model are largely different from reality (I would say wrong), we may stop at a completely wrong position. It will not be a question on how far it is from the correct position. It will stop at a completely wrong place.

Reviewers Response #2:

The authors compare the computation and processing time to a full model, but compare the performance of a simple model. The authors also suggest that a simple model can be catastrophic. There are many examples in the literature showing one-off machine learning predictions to be catastrophic, which is why machine learning manuscripts often show performance across an entire data set rather than a single sample. The authors have not shown that their method would not generate catastrophic results.

In summary, I am not satisfied with the manuscript in it's current form. Perhaps the authors are amicable to agreeing that this is a prototype concept tested on a single full model.

A suggested change to the title could be "GNSS Antenna Pattern Prediction and Placement Optimization: A prototype method using Machine Learning to Aid Complex Electromagnetic Simulations validated on a single sample"

Author Response

Authors Response #1

Thank you for your analysis. In fact, we decided to use two plots, not for hiding anything, but to not mix to many lines. We would like to keep it like that, which makes it easier to use it when printed. We agree that in this case, the deep learning is not better in all quadrants, but also not worst. And is better in the range [0-90º] and [90-135º]. This means that almost half the radiation diagram is improved

Reviewers Response:

In my opinion, half the radiation diagram improving for one sample is not indicative of the performance of the method as a whole. The choice of presenting the method or the results is inconsistent with contemporary Machine learning literature.

We understand that it would be much better if the prediction would fit completely with the full model. However, it may not look a great improvement, but for this type of simulations it’ s the difference between arriving at an antenna correct position, or not. Because the relevant part of the radiation is between -90, 90, which means radiation towards the sky. And since that the most relevant part, it is expectable that the model takes more attention to that region, because that is the part that will lead to a correct antenna localization at the end of the process. Actually, for our application (receive signal from satellite), we could even not consider the radiation going bellow the antenna. But we decided to include all the results, that may help others to see what is happening in all diagram regions. We would like also to enforce that the success of the methodology is not only if the aprox diagram is fully matching the full diagram, but if the final position indicated for antenna localization allows to obtain the required performance. And we observed that the final position fits our needs.

Authors Response #2:

The main benefit is time. Each simulation with the full model (for each candidate position) takes between one and two weeks (depends on how much we strip the vehicle elements). Depending on the simplification used, it was possible to make all simulations to get a position in less than one day. In some cases, in half day. The full model would take one or two months. This worth some penalty in accuracy. Actually, with this approach, we may select two or three candidate positions, and perform the full model simulation, to make the final decision, or perform anechoic chamber measurements.

The performance when using the simplified, sometimes, is catastrophic. Because it depends the path the localization algorithm uses. If it selects a path more on the middle of the roof, the reduced model follows it very well. But if it decides to go (or if it is required by design) around the roof, near the edges or corners, the start to become more different. And sometimes, since antenna characteristics from simplified model are largely different from reality (I would say wrong), we may stop at a completely wrong position. It will not be a question on how far it is from the correct position. It will stop at a completely wrong place.

Reviewers Response #2:

The authors compare the computation and processing time to a full model, but compare the performance of a simple model. The authors also suggest that a simple model can be catastrophic. There are many examples in the literature showing one-off machine learning predictions to be catastrophic, which is why machine learning manuscripts often show performance across an entire data set rather than a single sample. The authors have not shown that their method would not generate catastrophic results.

Dear reviewer, maybe we did not explain well, but the performance comparison is not with the simple model. Actually, figure 6-b) compares it with the full model. It would be, if we were not using the proposed methodology. What we intend to say is that it takes less (much less) time, with some potential penalty in accuracy, but that does not mean it is the accuracy of the simplified model. And in some cases, that penalty may be lower than what is tolerated by antenna specification. Also, we think that it’s acceptable to have different methods available, and then we can select the one that better fits our needs. In this case we will have three methods. The full model (that may take so long that it becomes useless), the simplified (that may become so inaccurate that is useless), and the proposed method, that takes less time, but keeps the accuracy at a useful level. This proposal is exactly to avoid the possible problems when only the simple model is inaccurate, and leads to catastrophic solutions. That does not happen with the proposed methodology. At least with the tested conditions. And we agree that we did not have a proof this is error free methodology. We mention that in the manuscript conclusions. But we have results that the methodology works for the explained conditions, and we think that reporting our results is a contribution for people working in the field, looking for solutions to reduce the simulation time for each iteration, as we were 2 years ago.

In summary, I am not satisfied with the manuscript in it's current form. Perhaps the authors are amicable to agreeing that this is a prototype concept tested on a single full model.

We already mention that in the conclusions. This methodology worked for our scenario, but before using it on other scenarios, users should verify its validity.

A suggested change to the title could be "GNSS Antenna Pattern Prediction and Placement Optimization: A prototype method using Machine Learning to Aid Complex Electromagnetic Simulations validated on a single sample"

It’s no common the title being suggested by reviewers, but we thanks your suggestion. However, “a single sample” does fully describes what is the sample. We propose:

"GNSS Antenna Pattern Prediction and Placement Optimization: A prototype method using Machine Learning to Aid Complex Electromagnetic Simulations validated on a vehicle model"

We hope our clarifications and title change is able to clarify your concerns.

Thank you very much.